# Rotor UAV’s Micro-Doppler Signal Detection and Parameter Estimation Based on FRFT-FSST

**DOI:** 10.3390/s21217314

**Published:** 2021-11-03

**Authors:** Huiling Hou, Zhiliang Yang, Cunsuo Pang

**Affiliations:** National Key Laboratory for Electronic Measurement Technology, North University of China, Taiyuan 030051, China; houhuiling@nuc.edu.cn (H.H.); yangzhiliang@nuc.edu.cn (Z.Y.)

**Keywords:** rotor UAV, FRFT, FSST, parameter estimation

## Abstract

The micro-Doppler signal generated by the rotors of an Unmanned Aerial Vehicle (UAV) contains the structural features and motion information of the target, which can be used for detection and classification of the target, however, the standard STFT has the problems such as the lower time-frequency resolution and larger error in rotor parameter estimation, an FRFT (Fractional Fourier Transform)-FSST (STFT based synchrosqueezing)-based method for micro-Doppler signal detection and parameter estimation is proposed in this paper. Firstly, the FRFT is used in the proposed method to eliminate the influence of the velocity and acceleration of the target on the time-frequency features of the echo signal from the rotors. Secondly, the higher time-frequency resolution of FSST is used to extract the time-frequency features of micro-Doppler signals. Moreover, the specific solution methodologies for the selection of window length in STFT and the estimation of rotor parameters are given in the proposed method. Finally, the effectiveness and accuracy of the proposed method for target detection and rotor parameter estimation are verified through simulation and measured data.

## 1. Introduction

In recent years, the incidents such as UAV crashing, injuring people, or damaging properties, etc. that civil UAVs endanger public safety have frequently occurred, for which the implementation of safety supervision and prevention has been a “new normal” in [1,2].

Since most of the civil UAVs have rotors, the micro-Doppler signals thus produced contain information such as the structural features and motion status of the target, which can be used to improve the accuracy of the classification and recognition of the target in [3,4]. For the analysis and recognition of the micro-Doppler in [5,6,7], the micro-Doppler features caused by the UAV rotors were studied, where Victor C. Chen pointed out that the micro-Doppler from the radial velocity was a very important feature for the target with multiple rotors while the micro-Doppler spectra caused by multiple rotors might affect each other, for which the high-precision time-frequency analysis techniques should be further studied. In [8], the micro-Doppler effect caused by small UAVs was studied, and the effectiveness of the micro-Doppler signal for target detection was verified by using simulation and measurement data. In [9], the STFT was used to analyze the parameters of the micro-Doppler signal and the window function was designed according to the instantaneous period of the data set, improving the time-frequency resolution of the signal. In [10], the FMCW radar was used to detect Nano-UAV (less than 5 cm), and the micro-Doppler features were extracted from experimental data. In [11], an IRNN-based algorithm for UAV detection and classification recognition was studied based on the micro-Doppler features with the recognition rate reaching more than 90%. In [12], a time-frequency analysis method for micro-Doppler signal based on STFRFT (Short-Time Fractional Fourier Transform) was studied, significantly improving the time-frequency resolution compared with STFT. In [13,14,15,16], the adaptive detection method of targets was studied in a clutter environment, which could be used to improve the detection performance of small UAV targets. In [17,18], a target detection method based on acoustic array signals was studied, and the experimental results indicated that it could be used as an effective means for low-altitude UAV detection. In [19,20,21], the automatic classification and detection methods of UAVs based on the micro-Doppler spectrum were studied, and the research indicated that the introduction of the neural network could improve the target recognition rate. In [22,23,24], a variety of methods for UAV detection and classification were studied, and the research indicated that the target recognition rate could be improved by introducing multi-modal and multi-dimensional deep learning methods. In [25,26,27], the time-frequency focus performance of the methods such as SST (Synchrosqueezing Transform), SST2 (Second-order Synchrosqueezing Transform), HSST (High-order Synchrosqueezing Transform), etc. to the time-varying signals was studied, which could be used to improve the time-frequency resolution of the micro-Doppler signal. In [28], it was pointed out that the micro-Doppler effect was an important feature for identifying small UAVs, and it was further proved that SST had a higher time-frequency resolution with measured data.

The above analysis indicates that the micro-Doppler signal plays an important role in the detectionand identification of UAVs, but the existing methods often fail to consider the influence of velocity and acceleration on the micro-Doppler signal. Meanwhile, when the number of rotors increases, the time-frequency resolution by the STFT method is too low to accurately judge the rotor parameters. To this end, starting from the influence of velocity, acceleration, and multi-rotor parameters on the time-frequency features of the micro-Doppler signal, the FRFT-based method for velocity and acceleration estimation has been studied in this paper, together with the study on the method for extraction of the multi-component micro-Doppler signal features based on FSST, combined with the feature of higher time-frequency resolution of SST, and the high-precision estimation of rotor parameters has been realized.

An echo model of the rotor UAV is given in the Section 2 of this paper; a proposed algorithm is given in the Section 3; a method for the parameter estimation of the proposed algorithm is indicated in the Section 4; the influence of the rotor parameters on the focus performance of micro-Doppler signal is analyzed in the Section 5; the effectiveness of the algorithm is verified by using simulation and measured data in the Section 6.

## 2. Echo Signal Model

### 2.1. Signal Model

The geometric configuration relationship between the UAV and the radar is shown in Figure 1, where the radar is located at the origin of the coordinate system (X,Y), the reference coordinate system (x,y) is the translation of the radar coordinate system, whose origin O is the geometric center of the UAV. In order to simplify the analysis, here, it is assumed that the azimuth angle and the elevation angle of the UAV relative to the radar are both 0.

Suppose $P_{0}$ is a scattering point on the rotor, which rotates around the center point o, with the rotation frequency being $f_{r}$. The distance from $P_{0}$ to point o is l, and the distance from radar to point o is $R_{0}$, where $R_{0}$ is a function of time *t*. Here, let *t* = 0, the initial rotation angle of $P_{0}$ is $\theta_{0}$, then point $P_{0}$ rotates to point $P_{t}$ when going through time *t*. Since it is assumed that the radar and the rotor are on the same plane (x,y,z=0), the distance from point $P_{t}$ to the radar can be expressed as [4,29]:(1)$$\begin{array}{cc} R_{P}(t) & ={\left[R_{0}^{2}+l^{2}+2lR_{0}\sin(\theta_{0}+2\pi f_{r}t)\right]}^{1/2}\\ & \approx R_{0}+V_{R}t+l\sin\theta_{0}\cos2\pi f_{r}t+l\cos\theta_{0}\sin2\pi f_{r}t \end{array}$$
where, $V_{R}=v_{0}+at$ is the radial velocity of the UAV, $v_{0}$ is the initial velocity, and a is the acceleration, usually ${\left(l/R_{0}\right)}^{2}\to 0$ in the far field. Therefore, the echo signal of the scattering point $P_{t}$ received by the radar can be simplified as:(2)$$\begin{array}{l} s_{R}(t)=\delta_{m}\exp\{j[2\pi f_{0}t+\Phi_{P}(t)]\}\\ =\delta_{m}\exp\{j[2\pi f_{0}t+\frac{4\pi }{\lambda }R_{P}(t)]\}\\ =\delta_{m}\exp\{j[2\pi f_{0}t+\frac{4\pi }{\lambda }(R_{0}+V_{R}t+l\sin\theta_{0}\cos2\pi f_{r}t+l\cos\theta_{0}\sin2\pi f_{r}t)]\} \end{array}$$
where, $\delta_{m}$ is the reflection coefficient of the rotor and the phase function is $\Phi_{P}(t)=4\pi R_{P}(t)/\lambda$.

Performing down-conversion processing on (2) and ignoring the influence of the constant term, the baseband echo signal of scattering point $P_{t}$ can be obtained as:(3)$$s_{B}(t)=\delta_{m}\exp(j\frac{4\pi }{\lambda }V_{R}t)\exp\{j\frac{4\pi }{\lambda }l\sin(2\pi f_{r}t+\theta_{0})\}$$

Suppose the length of the blade of the rotor is *L*, when it is integrated according to (3), the baseband echo signal of the entire blade is:(4)$$\begin{array}{l} s_{L}(t)=\delta_{m}\exp(j\frac{4\pi }{\lambda }V_{R}t)\int_{l=0}^{L}\exp\{j\frac{4\pi }{\lambda }l\sin(2\pi f_{r}t+\theta_{0})\}\\ =\delta_{m}L\exp(j\frac{4\pi }{\lambda }V_{R}t)\exp\{j\frac{4\pi }{\lambda }\frac{L}{2}\sin(2\pi f_{r}t+\theta_{0})\}\sin c\{\frac{4\pi }{\lambda }\frac{L}{2}\sin(2\pi f_{r}t+\theta_{0})\} \end{array}$$

According to (4), the total received signal of Z blades can be expressed as:(5)$$\begin{array}{l} s_{\Sigma }(t)=\sum_{k=0}^{Z-1}s_{Lk}(t)\\ =\exp(j\frac{4\pi }{\lambda }V_{R}t)\sum_{k=0}^{Z-1}\delta_{m}L\sin c[\phi_{k}(t)]\exp\{j\phi_{k}(t)\} \end{array}$$
where, $\theta_{k}=\theta_{0}+2k\pi /Z$(k=0,1,2⋯Z−1) the initial rotation angle of *Z* different blades, $\phi_{k}(t)$ represents its phase function, expressed as:(6)$$\phi_{k}(t)=\frac{4\pi }{\lambda }V_{R}t+\frac{4\pi }{\lambda }\frac{L}{2}\sin(2\pi f_{r}t+\theta_{k})$$

When β≠0, Formula (5) can be turned as:(7)$$s_{\Sigma }(t)=\exp(j\frac{4\pi }{\lambda }V_{R}t)\sum_{k=0}^{Z-1}\delta_{m}L\sin c[\phi \prime_{k}(t)]\exp\{j\phi \prime_{k}(t)\}$$
where, $\phi \prime_{k}(t)=\frac{4\pi }{\lambda }V_{R}t+\frac{4\pi }{\lambda }\frac{L}{2}\cos\beta \sin(2\pi f_{r}t+\theta_{k})$.

When considering the echo signal from the body, the echo signal of the UAV can be obtained as:(8)$$S(t)=S_{D}(t)+S_{m}(t)$$
where, $S_{m}(t)=s_{\Sigma }(t)$ is the echo signal of the rotor, and $S_{D}(t)$ is the echo signal of the body, which can be expressed as:(9)$$S_{D}(t)=\delta_{D}\exp(-j\frac{4\pi }{\lambda }V_{R}t)$$

### 2.2. Doppler Features of UAV

Performing derivative operation for the phase function of $S_{D}(t)$ in (8), and the Doppler frequency caused by the body can be obtained as:(10)$$f_{D}(t)=\frac{2}{\lambda }(v_{0}+at)$$

Performing derivative operation for the phase function of $S_{m}(t)$ in (8), and the Doppler frequency caused by the kth rotor can be obtained as:(11)$$f_{km}(t)=f_{D}(t)+\frac{2\pi }{\lambda }[\cos\beta Lf_{r}(-\sin\theta_{0}\sin2\pi f_{r}t+\cos\theta_{0}\cos2\pi f_{r}t)]$$

Formula (11) shows that the micro-Doppler frequency of the signal is modulated by the rotation rate $f_{r}$ through the functions $\sin2\pi f_{r}t$ and $\cos2\pi f_{r}t$.

## 3. Micro-Doppler Signal Analysis Based on FRFT-FSST

### 3.1. Velocity and Acceleration Compensation Based on FRFT

It can be seen from (10) and (11) that velocity and acceleration will cause the translation and tilt of the time-frequency curve time-frequency, even folding in severe cases, making it impossible to estimate the relevant parameters. The velocity and acceleration would cause the phase of the target echo to turn into the approximate form of a linear FM signal, and the FRFT has advantages in LFM parameter detection. Therefore, the FRFT has been used to estimate the velocity and acceleration, and further construct the penalty function so as to eliminate its influence on the time-frequency features of the echo.

FRFT is performed to deal with (8) to obtain [30]:(12)$$X_{p}(u)=\int_{-\infty }^{+\infty }S(t)K_{p}(t,u)dt$$

Kernel function $K_{p}\left(u,t\right)$ is [31];
(13)$$K_{p}\left(t,u\right)=\{\begin{array}{c} A_{p}\exp\left[j\pi \left(t^{2}+u^{2}\right)\cot\alpha -j2\pi tu\csc\alpha \right],\;\alpha \neq k\pi \\ \;\;\;\;\;\;\delta \left(u-t\right),\;\;\;\;\;\;\;\;\alpha =2k\pi \\ \;\;\;\;\;\;\delta \left(u+t\right),\;\;\;\;\;\;\alpha =\left(2k+1\right)\pi \end{array}$$
where, $A_{p}=\exp\left[-j\pi \mathrm{sgn}\left(\sin\alpha \right)/4+j\alpha /2\right]/\sqrt{\left|\sin\alpha \right|}$, k∈ℤ, α=pπ/2.

Maximum value search is preformed for (12) to get the estimated value of velocity and acceleration [32,33]:(14)$${\hat{v}}_{0}=\frac{\lambda \hat{u}\csc(\hat{p}\pi /2)}{2}$$
(15)$$\hat{a}=-\frac{\lambda \cot(\hat{p}\pi /2)}{2}$$
where, $\left(\hat{p},\hat{u}\right)$ can be estimated with Formula (16), that is:(16)$$(\hat{u},\hat{p})=\underset{p}{\arg\max}{\left|X_{p}(u))\right|}^{2}$$

Formulas (14) and (15) are utilized to construct the echo compensation function $L_{D}(t)$, and Formula (8) is used to obtain the compensated function as:(17)$$\begin{array}{cc} S_{L}(t) & =S(t)L_{D}(t)\\ & =S(t)\exp[j\frac{4\pi }{\lambda }({\hat{v}}_{0}+\hat{a}t)t] \end{array}$$

When compensated by (17), the influence of velocity and acceleration on the time-frequency curve has been basically eliminated.

Example 1: Suppose that the radar carrier frequency is 5 GHz, the baseband sampling rate 10 KHz, the sampling time 1 s, the number of target blades 2, the rotation frequency 2 Hz/s, the length 6 m, the elevation angle of the blades relative to the radar $\beta =0^{0}$, not considering the body echo in the analysis, the signal-to-noise ratio of the blade echo is −3 dB. Figure 2a,b indicate the influence of velocity and acceleration on the time-frequency curve; Figure 2c shows the time-frequency curve when processed by FRFT compensation (at velocity of 19.6 m/s, acceleration of 60.4 m/s^2^). It can be seen from Figure 2 that when the velocity and acceleration increase, the time-frequency curve will be folded, which will lead to an increase of errors in the subsequent parameter estimation. When the compensation process is carried out by using (17), the influence of the velocity and acceleration on the accuracy of parameter estimation can be eliminated, as shown in Figure 2c.

### 3.2. Time-Frequency Analysis Based on FSST

STFT had been widely used in the analysis of micro-Doppler signals [34,35], but the selection of window length was very important for the analysis of echo signals at different rotation frequencies; therefore, a selection method is presented in this paper by the experiential knowledge to realize the adaptive selection of window length; at the same time, in order to further improve the aggregation performance of the time-frequency curve, the SST method is introduced to improve the time-frequency resolution of the signal.

STFT is performed to (17) to obtain:(18)$$Y(\tau ,w)=\int S_{L}(t)g(t-\tau )e^{-jwt}dt$$
where, $g(t)=\exp(-c\frac{t^{2}}{2})$ is the Gaussian function, $c=1/\delta_{t}^{2}$, $\delta_{t}$ represents 3 dB width of Gauss signal in time domain.

The time-frequency resolution performance in (18) is directly related to the selection of window length. Here, according to the requirement that the micro-Doppler bandwidth of the signal to be analyzed and the length of the analyzed signal shall be less than a quarter period, the selected window length $g_{{}_{W}}$ is as:(19)$$g_{W}=\min[\frac{20F}{f_{mBW}},\frac{1}{4}\frac{F}{f_{r}}]$$
where, *F* is the signal sampling frequency, and $f_{mBW}$ is the micro-Doppler spectrum bandwidth caused by the tip of the blade, it can be expressed from the (11) as follows:(20)$$f_{m}{}_{BW}=\frac{8\pi f_{r}L}{\lambda }\cos\beta$$

In order to further improve the time-frequency concentration performance of STFT, SST is used in this section for subsequent processing.

Performing SST on Formula (18), the time-frequency diagram after rearrangement can be expressed as [25,36]:(21)$$T_{Y}(\tau ,w)=\frac{1}{g(0)}\int \delta [w-{\hat{w}}_{f}(\tau ,w)]Y(\tau ,w)dw$$
where, ${\hat{w}}_{f}(\tau ,w)$ is the instantaneous frequency, which can be expressed as
(22)$${\hat{w}}_{f}(\tau ,w)=-\frac{1}{jY(\tau ,w)}\frac{\partial Y(\tau ,w)}{\partial w}$$

It can be seen by comparing Equations (21) and (18) that in Equation (21), the time-frequency focus processing has been performed, which is conducive to the time-frequency resolution and parameter estimation of multi-component micro-Doppler signals.

## 4. Parameter Estimation

In order to simplify the analysis, assuming that the velocity and acceleration have been compensated, the form of the numerical dispersion of (17) is rewritten as:(23)$$S_{L}(nT_{s})=S_{LD}(nT_{s})+S_{Lm}(nT_{s})$$
where, n=0,1,2⋯N is the number of signal sampling points, and $T_{s}=1/F$ is the interval of signal sampling time.

### 4.1. Estimation of the Number of Rotors

The multiple time points $\tau_{Q}$ on the time-frequency image $T_{Y}[\tau ,w]$ in (21) are randomly extracted to estimate $\hat{z}$, the number of UAV rotors, namely:(24)$$\hat{z}=\frac{1}{\tau_{Q}}\sum_{i=1}^{\tau_{Q}}q_{i}$$
where, $q_{i}$ represents the sum of the number of points with the amplitude greater than the threshold in the extracted ith point, which can be expressed as:(25)$$q_{i}=\sum \left\{\left|\left[T_{Y}(\tau_{Q},w)\right]\right|>\eta \max[T_{Y}(\tau ,w)]\right\},i\in [1,\tau_{Q}],\tau_{Q}<<N$$
where η∈[0.6,0.9] is the threshold coefficient. The higher coefficient is generally suitable for high SNR signals. The lower coefficient is conducive to the detection of low SNR signals, but it will also lead to the increase of false targets.

### 4.2. Estimation of Rotation Frequency $f_{r}$

When performing FFT processing on (17), the frequency interval Δf of adjacent spectral lines is obtained, and the rotation frequency $f_{r}$ is obtained according to Δf:(26)$${\hat{f}}_{r}=\frac{\Delta f}{\hat{z}}=\frac{\left(n_{a+1}-n_{a}\right)}{\hat{z}NT_{s}}$$
where, $n_{a}$, $n_{a+1}$ is the number of frequency points corresponding to the adjacent spectral line of the maximum peak point in (17).

In addition, by performing FFT on $T_{Y}[\tau ,w]$ by the time dimension in (21), the rotation frequency $f_{r}$ can also be obtained, namely:(27)$${\hat{f}}_{r}=\frac{\left(n\prime_{a+1}-n\prime_{a}\right)}{\hat{z}N\prime T_{s}}$$
where, N′ is the point number of signal analysis, $n\prime_{a}$ and $n\prime_{a+1}$ are respectively the frequency points corresponding to the main lobe and the first side lobe in $FFT{\left\{T_{Y}[\tau ,w]\right\}}_{\tau }$ spectrum.

### 4.3. Estimation of Rotor Length $\hat{L}$

According to (20), the expression of $\hat{L}$ can be obtained as [37]:(28)$$\hat{L}=\frac{\lambda {\hat{f}}_{mBW}}{8\pi f_{r}\cos\beta }$$
where, the numerical expression of ${\hat{f}}_{mBW}$ is
(29)$${\hat{f}}_{mBW}=\frac{a_{0+}-a_{0-}}{NT_{s}}$$
where, $a_{0+},a_{0-}$ represents the number of frequency points corresponding to the maximum value on the left and right sides of the origin in $FFT\{S_{L}(nT_{s})\}$ spectrum.

Figure 3 shows the parameter estimation process of the proposed algorithm.The steps of the proposed algorithm are as follows:
(1)FRFT is performed on the echo signal to estimate the velocity and acceleration of the target, and then a penalty function is constructed to eliminate the influence of the velocity and acceleration on the time-frequency curve to obtain the function $S_{L}(t)$;(2)FFT is performed on $S_{L}(t)$ to obtain Δf,the interval of adjacent spectral lines, ${\hat{f}}_{mBW}$, the micro-Doppler bandwidth, and then $g_{W}$,the estimated window length is given;(3)$g_{W}$ is used to perform STFT and SST processing on the echo signal to obtain a high-resolution time-spectrogram;(4)According to the time-frequency time-spectrogram, Equations (24), (26), and (28) are used to estimate the number of rotors, rotation frequency, and rotor length.

## 5. Influence of Rotor Parameters on Time-Frequency Features

### 5.1. Influence of Rotor Number and Rotation Frequency on Time-Frequency Features

Suppose that the simulation parameters are the same as those in Example 1. Figure 4 shows the results of the proposed algorithm when the number of rotors is 2 and 4 respectively, of which Table 1 shows the configuration of each parameter in Figure 4. Figure 4a,b indicate that the time-frequency resolution will not decrease as the number of rotors increases when the rotation frequency is 2 and the number of rotors is changed. Figure 4a,c show that the time-frequency resolution will be significantly decreased when the number of rotors is 2, the rotation frequency is changed, and the window length remains the same. Figure 4c,d show that the time-frequency resolution will be significantly improved when the rotation frequency is doubled compared to Figure 4a and the window length is reduced. The above results indicate that the time-frequency resolution of the proposed algorithm is mainly related to the window length of the window function determined by the rotation frequency. The larger the rotation frequency, the shorter the window length, and vice versa.

### 5.2. Influence of Time-Varying Rotation Frequency on Time-Frequency Features

Figure 4 and Table 1 in Section 5.1 show that when $\hat{z}$ and $f_{r}$ are fixed, it can obtain better time-frequency resolution to use $g_{W}$ as the window length of the FSST; but in the actual UAV echo signal processing, $f_{r}$ may vary as $\Delta f_{r}$ does, for which the adaptability of the proposed algorithm has been analyzed under different $\Delta f_{r}$. Figure 5 shows the processing results of the proposed algorithm when $\Delta f_{r}$ is at 1 Hz/s and 2 Hz/s, respectively, of which Table 2 shows the parameter configuration used in each sub-graph in Figure 5. Comparing Figure 5a,b, it can be seen that as $\Delta f_{r}$ increases, the time-frequency focus performance of the proposed algorithm becomes worse.

In order to solve the above problems, Equation (29) can be utilized to estimate ${\hat{f}}_{m}{}_{BW}$ for the signal segmentation. Figure 6 gives out the FFT processing results of different segmented signals when $\Delta f_{r}$ is at 1 Hz/s and 2 Hz/s respectively; meanwhile, Table 3 gives out ${\hat{f}}_{m}{}_{BW}$ corresponding to each sub-graph in Figure 6, from which it can be seen that the larger $\Delta f_{r}$ is, the wider the broadening range of ${\hat{f}}_{m}{}_{BW}$ is, and the larger the variation range of $g_{W}$ is. Therefore, for the echo signals with variable rotation frequencies, no fixed window length can be used for time-frequency matching. Figure 7 shows the processing results of the proposed algorithm when $\Delta f_{r}$ is at 2 Hz/s, of which Items (c) and (d) in Table 3 are the parameter configurations used in Figure 7.

Comparing Figure 5b and Figure 7, it can be seen that the time-frequency focus performance of the proposed method is obviously improved when the window length $g_{W}$ is selected by segmentation; the above results show that for the echo signals with variable rotation frequencies, it can improve the time-frequency resolution of the signal when the method for signal segmentation is used to select the window length. In addition, further in-depth research shall be conducted on how to reasonably segment the signal.

## 6. Analysis on Algorithm Performance

In this section, the simulation data and measured data are used for verification in order to further validate the effectiveness and adaptability of the proposed algorithm, of which the simulation data are the small UAV echo signals collected by a W-band radar, which are used in the experiment to verify, on the one hand, the higher time-frequency resolution of the proposed algorithm than that of STFT, and, on the other hand, the correctness of the proposed algorithm to the parameter estimation such as the parameter compensation, window length selection $g_{W}$, micro-Doppler bandwidth ${\hat{f}}_{mBW}$, and rotation frequency; and the measured data are the echo data from a model helicopter collected by a P-band radar, which are used in the experiment to verify, on the one hand, the micro-Doppler features of the UAV under low-frequency radar detection, and on the other hand, it can be seen from Equation (20) that the micro-Doppler frequency bandwidth is proportional to the length of the rotor and the radar frequency, which means increasing the length of the rotor can obtain the micro-Doppler signal equivalent to that generated with the high-frequency radar detecting UAV, based on which to further verify the performance of the proposed algorithm in terms of target detection and parameter estimation.

### 6.1. Simulation Experiment

Assuming that the radar is provided with a CW Regime, the radar parameters and target parameters are shown in Table 4; the signal-to-noise ratio(SNR) of the UAV fuselage is 5 dB higher than that of the rotor echo, and the SNR ratio between the rotor echo and Gaussian noise is 5 dB. The results of the proposed algorithm processed according to the flow path in Figure 3 are shown in Figure 8, of which Figure 8a is the results from FFT, it can be obtained by Equation (29) that ${\hat{f}}_{mBW}$ is at about 20 KHz, quite different from 4 KHz, the theoretical value calculated by Equation (20), and the experimental results indicate that the existence of the velocity and acceleration of the target has a greater influence on the estimation of micro-Doppler bandwidth, especially the spectrum broadening caused by acceleration has an even greater influence on the estimation of ${\hat{f}}_{mBW}$; Figure 8b is the results from performing FRFT compensation processing (with the velocity at 4.97 m/s, and the acceleration at 59.96 m/s^2^), it can be obtained by Equation (29) that ${\hat{f}}_{mBW}$ is approximately at 3.88 KHz, basically consistent with the theoretical value of 4 KHz, verifying the effectiveness of FRFT compensation processing; Figure 8c is the amplified results of Figure 8b, from which it can be obtained that the adjacent frequency spectrum interval is 40 Hz. Figure 8d is the STFT processing results of Figure 8b, showing that the time-frequency focus performance is poor; Figure 8e is the FSST processing results of Figure 8b, whose time-frequency focus performance has been greatly improved compared with that of Figure 8d; Figure 8f is the estimated result of the number of rotors in Figure 8b when processed by Equation (20), showing that the number of rotors is 4; based on the above results, combined with Figure 8c (with adjacent spectrum interval at 40 Hz), Equations (27) and (28), it can be obtained that the rotation frequency is 10 Hz/s and the rotor length is 9.7 cm, basically consistent with the theoretical value, and verifying the correctness of the parameter estimation of the established algorithm; Figure 8g is the FSST results without velocity and acceleration compensation processing, whose time-frequency curve has a serious deviation compared with Figure 8e, making it difficult to complete the parameter estimation accurately.

### 6.2. Measured Data

In order to verify the performance of the algorithm for the detection and parameter estimation of the actual target, a P-band radar is used in the experiment to conduct data acquisition and algorithm verification on the model helicopter. The system configuration is shown in Figure 9, and the system parameters are shown in Table 5.

For the echo signals in Figure 9, the processing results of the proposed algorithm are displayed in Figure 10. Figure 10a is the FFT of the echo signal, from which it can be seen that the SNR of the target fuselage is 5 dB higher than that of the rotor. Figure 10b is the FFT results of Figure 10a when the fuselage echo is removed by using the CLEAN [38,39], that’s because, usually, the echo signal noise of the fuselage is relatively large, and the fuselage echo can be removed by the CLEAN method so as to facilitate the subsequent analysis of the micro-Doppler signal. In Figure 10b, ${\hat{f}}_{mBW}\approx 950\;\mathrm{Hz}$ and Δf≈75 Hz can be obtained by using Equations (26) and (29); Figure 10c is the STFT results (wherein the window length $g_{W}$ obtained according to Equation (19) is 40), indicating that the time-frequency curves produced by the two rotors have poorer focus performance; Figure 10d is the FSST-processed results of the proposed algorithm, showing that the time-frequency focus is better than that in Figure 10c, and the number of rotors is 2. Substituting the above parameters into Equations (26) and (28), it can be obtained that the rotation frequency is about 37.5 Hz, and the blade length is about 680 mm, basically consistent with the actual values, and verifying the correctness of the parameter estimation by the proposed algorithm.

### 6.3. Analysis of Algorithm Complexity

Assuming that the signal length is N, the window width is $g_{{}_{W}}$, the compensation number of the fractional-order is k′, the complex multiplication operational quantity of standard STFT in Equation (18) is $O[\frac{N}{2}\log_{2}(g_{w})]$, the complex multiplication operational quantity of FRFT in Equation (12) is $O[\frac{N}{2}k\prime \log_{2}(N)]$, the operational quantity of SST in Equation (21) is $O[g_{W}N]$, and the total complex multiplication operational quantity of the proposed algorithm is approximately $O[\frac{N}{2}k\prime \log_{2}(N)+g_{W}N]$, taking Section 6.1 as an example, the complexity of the proposed algorithm is about one order of magnitude higher than that of standard STFT while improving performance.

## 7. Conclusions

In this paper, the echo model of the rotor UAV was analyzed and FRFT was used to perform the compensation processing on the velocity and acceleration of the target, which improved the estimation accuracy of the micro-Doppler signal parameters of the rotors. Simultaneously, FSST was used to carry out the time-frequency analysis on the rotor echo signal, the method to estimate the parameters such as the rotor rotation frequency, rotor number, and rotor length was given out, and the empirical Formula was given for the selection of the window length of Gaussian window function, which improved the time-frequency resolution and operation efficiency of STFT. Simulation and experimental data proved that the proposed method could accurately estimate the motion parameters of the UAV’s rotors. Relevant research results can be applied to the detection, classification, and recognition of the low-altitude UAVs; in addition, for the signals with further reduced SNR, the algorithm performance can be improved combined with deep learning methods.

## Figures and Tables

**Figure 1 sensors-21-07314-f001:**
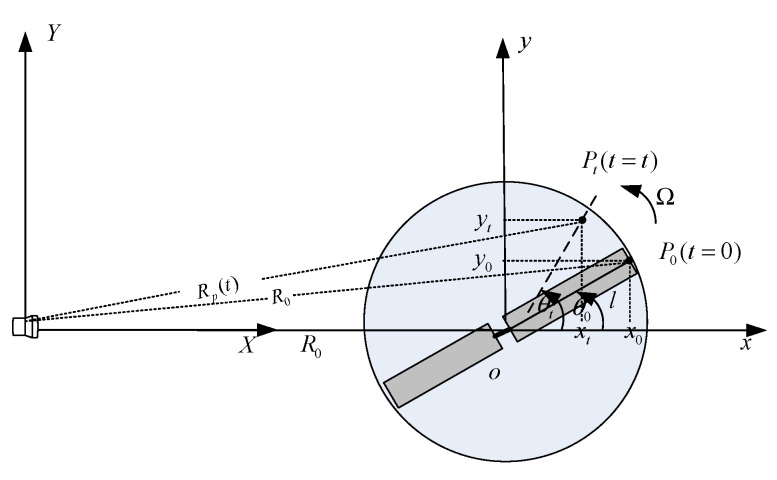
Geometric configuration between radar and UAV rotors.

**Figure 2 sensors-21-07314-f002:**
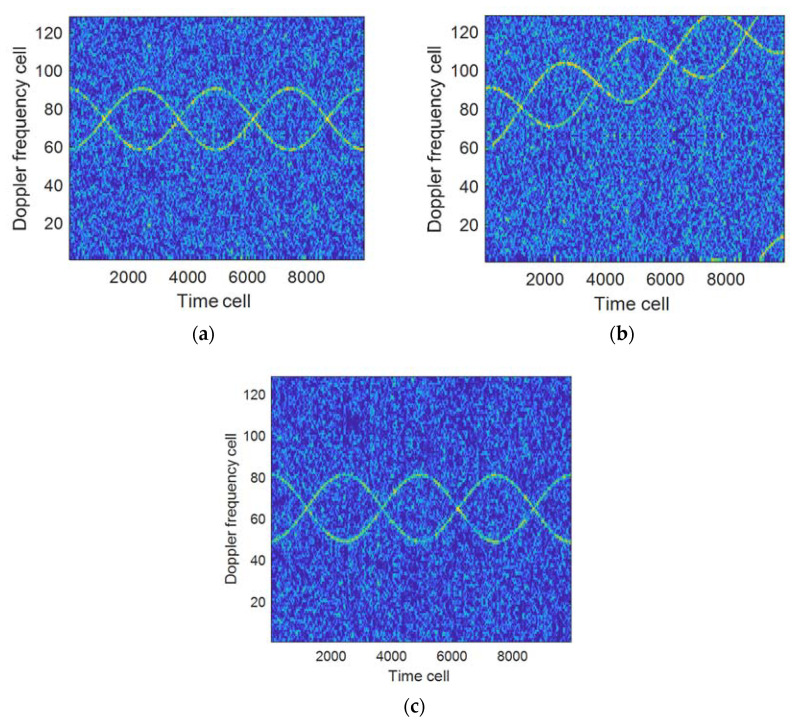
The influence of velocity and acceleration on the time-frequency curve. (**a**) ${\hat{v}}_{0}=20\;m/s,\;a=0\;m/s^{2}$; (**b**) ${\hat{v}}_{0}=20\;m/s,\;a=60\;m/s^{2}$; (**c**) Results when compensated (${\hat{v}}_{0}=19.6\;m/s,\;a=60.4\;m/s^{2}$).

**Figure 3 sensors-21-07314-f003:**
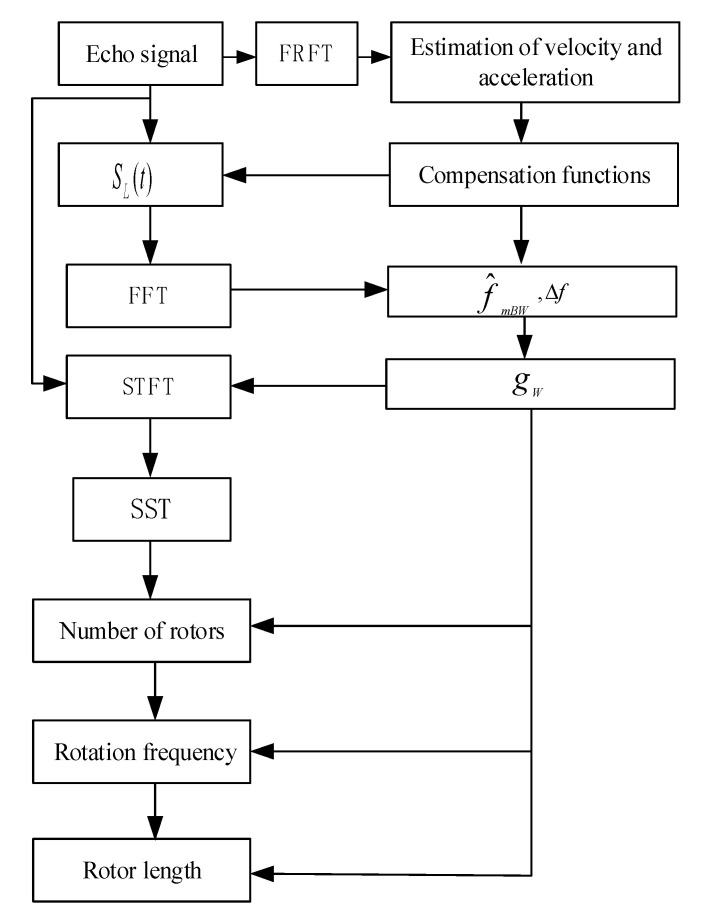
The processing flow of proposed algorithm.

**Figure 4 sensors-21-07314-f004:**
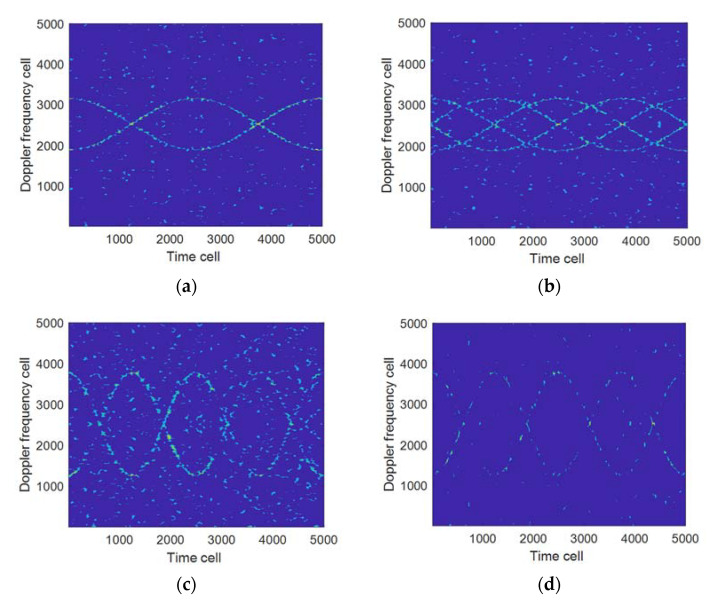
The influence of different numbers of rotors and different rotation frequencies on the time-frequency resolution. (**a**) $\hat{z}=2,f_{r}=2,g_{{}_{W}}=80$; (**b**) $\hat{z}=4,f_{r}=2,g_{{}_{W}}=80$; (**c**) $\hat{z}=2,f_{r}=4,g_{{}_{W}}=80$; (**d**) $\hat{z}=2,f_{r}=4,g_{{}_{W}}=40$.

**Figure 5 sensors-21-07314-f005:**
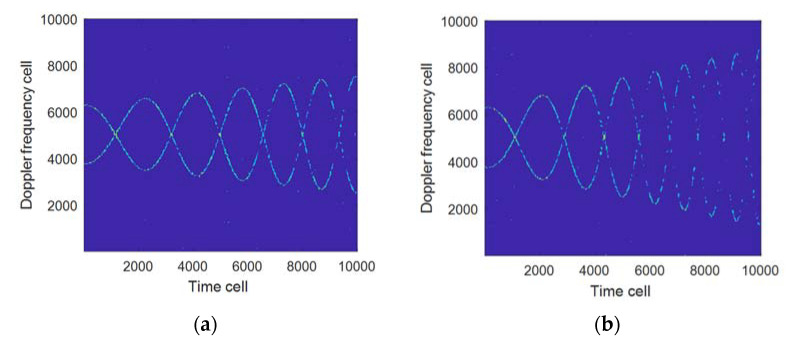
Processed results when rotation frequency changes. (**a**) When rotation frequency variation is 1 Hz/s ($g_{{}_{W}\;=\;80}$); (**b**) When rotation frequency variation is 2 Hz/s ($g_{{}_{W}\;=\;80}$).

**Figure 6 sensors-21-07314-f006:**
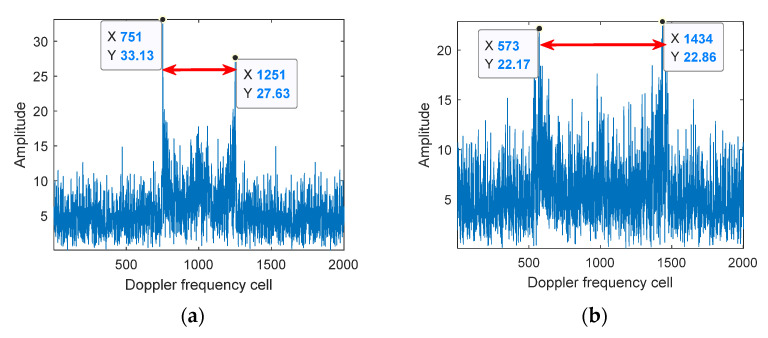
Micro-Doppler spectrum broadening variation when the rotation frequency changes. (**a**) ${\hat{f}}_{mBW}$ ($\Delta f_{r}=1\;\mathrm{Hz}/s$, first 2000 points); (**b**) ${\hat{f}}_{mBW}$ ($\Delta f_{r}=1\;\mathrm{Hz}/s$, last 2000 points); (**c**) ${\hat{f}}_{mBW}$ ($\Delta f_{r}=2\;\mathrm{Hz}/s$, first 2000 points); (**d**) ${\hat{f}}_{mBW}$ ($\Delta f_{r}=2\;\mathrm{Hz}/s$, last 2000 points).

**Figure 7 sensors-21-07314-f007:**
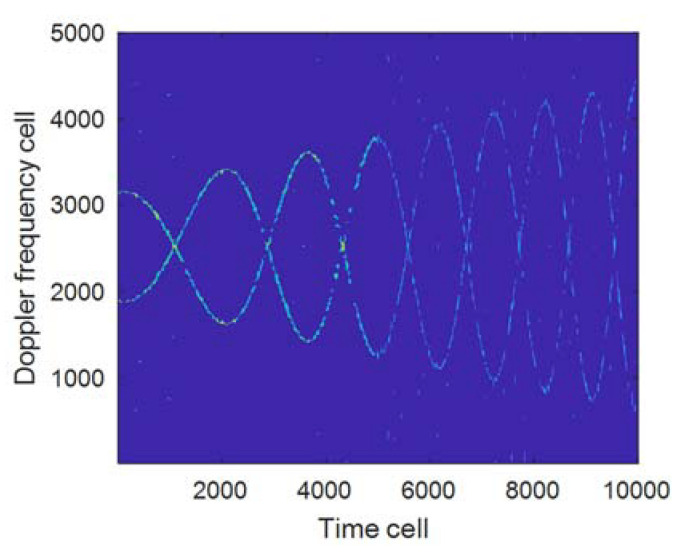
Time-frequency results corresponding to $g_{{}_{W}}$ by segment ($g_{{}_{W1}\;=\;80},g_{{}_{W2}\;=\;38}$).

**Figure 8 sensors-21-07314-f008:**
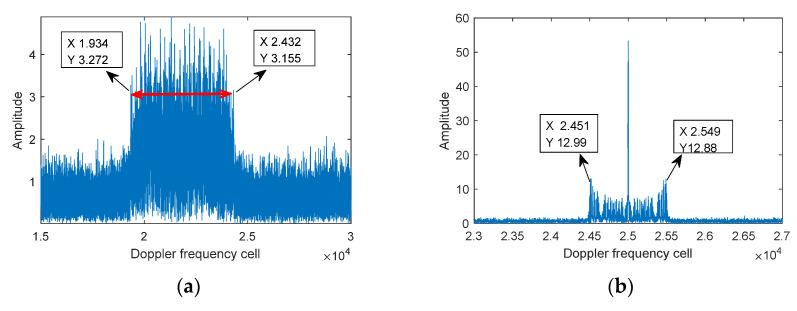
The processed results of proposed algorithm. (**a**) FFT; (**b**) FFT with parameter compensated; (**c**) Δf=40 Hz; (**d**) STFT ($g_{{}_{W}}=1000$); (**e**) FRFT-FSST; (**f**) Estimation of rotor number; (**g**) FSST result without compensation processing.

**Figure 9 sensors-21-07314-f009:**
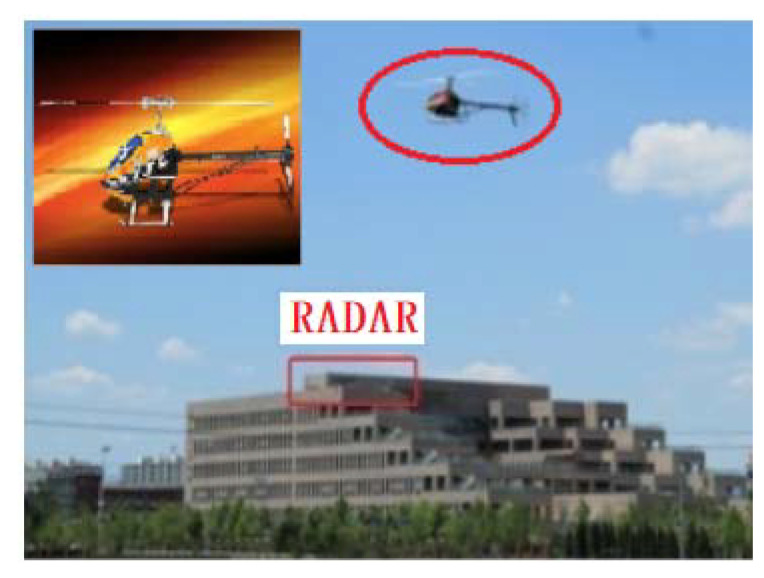
Data acquisition system.

**Figure 10 sensors-21-07314-f010:**
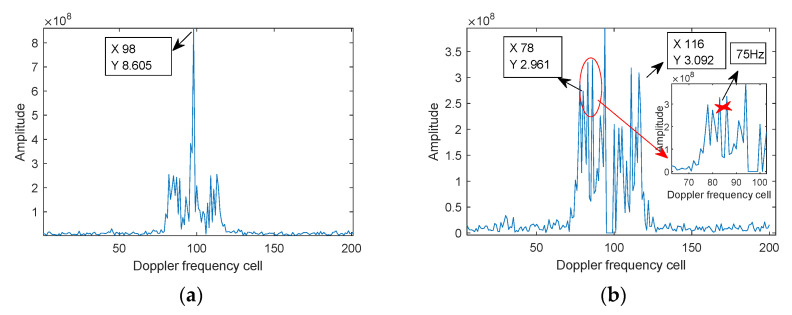
Data processing of helicopter. (**a**) FFT, (**b**) analysis of ${\hat{f}}_{m}{}_{BW},\Delta f$, (**c**) STFT, (**d**) FSST and rotor number analysis.

**Table 1 sensors-21-07314-t001:** Configuration of each parameter in Figure 4.

Figure 4	$\hat{z}$	$f_{r}$	${\hat{f}}_{mBW}$	$g_{W}$ (Theoretical)	$g_{W}$ (Practical)
(a)	2	2	2.5 KHz	80	80
(b)	4	2	2.5 KHz	80	80
(c)	2	4	5 KHz	40	80
(d)	2	4	5 KHz	40	40

Note: $g_{W}$ is obtained according to Equation (19).

**Table 2 sensors-21-07314-t002:** Configuration of each parameter in Figure 5.

Figure 5	$\hat{z}$	$f_{r}$	$\Delta f_{r}$	${\hat{f}}_{mBW}$	$g_{W}$ (Theoretical)	$g_{W}$ (Practical)
(a)	2	2	1 Hz/s	2.5 KHz	80	80
(b)	2	2	2 Hz/s	2.5 KHz	80	80

Note: $g_{W}$ is obtained according to Equation (19), where ${\hat{f}}_{mBW}$ is calculated by $f_{r}=2$.

**Table 3 sensors-21-07314-t003:** Calculation of each parameter in Figure 6.

Figure 6	$\hat{z}$	$f_{r}$	$\Delta f_{r}$	${\hat{f}}_{mBW}$ (Front 2000 Points)	${\hat{f}}_{mBW}$ (Back 2000 Points)	$g_{W}$ (Practical)
(a)	2	2	1 Hz/s	2.5 KHz		80
(b)	2	2	1 Hz/s		4.3 KHz	46
(c)	2	2	2 Hz/s	2.5 KHz		80
(d)	2	2	2 Hz/s		5.3 KHz	38

**Table 4 sensors-21-07314-t004:** Radar and Target Parameters.

Radar Parameters	Carrier Frequency	Sampling Frequency	Sampling Time	β	
	95 GHz	200 KHz	250 ms	0°	
Target parameters	Number of rotors	Rotation frequency	Rotor length	Velocity	Acceleration
	4	10 Hz/s	10 cm	5 m/s	60 m/s^2^

**Table 5 sensors-21-07314-t005:** Radar and Target Parameters.

Radar Parameters	Carrier Frequency	Sampling Frequency	Sampling Time	β	
	674 MHz	5 KHz	150 ms	40~60°	
Target parameters	Number of rotors	Rotation frequency	Rotor length	Velocity	Acceleration
	2	26~42 Hz/s	700 mm	<3 m/s	<1 m/s^2^

## Data Availability

The data presented in this study are available on request from the corresponding author.

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
