# Peer review of "Rotor UAV’s Micro-Doppler Signal Detection and Parameter Estimation Based on FRFT-FSST"

_sensors, 2021, doi:10.3390/s21217314_

Round 1
Reviewer 1 Report
- There is some strange formatting in the abstract on the line with FRFT-FSST that makes the line difficult to read.
- Many of the sentences in the abstract are disjointed or run-on, making it difficult to read, but all of the important information is there.
- The first sentence in the introduction is too long.
- Many of the sentences in the second paragraph of the introduction are too long and introduce too many concepts or citations at once.
- The literature review in the introduction is sufficient, but you have not clearly established what contributions you make in this paper compared to the existing literature. It looks like you have a short paragraph that claims you are studying FRFT and SST-STFT, but I would like to see a little bit more context about the perceived shortcomings in the previous literature and what advantages your proposed method(s) offer.
- Throughout the manuscript there are both missing spaces and extra spaces in the text.
- If you use equations as part of a sentence, consider not starting a new paragraph immediately afterward and not capitalizing the first letter (for example, after end (1) with a comma followed by "where V_R = v_0 + at" with no indent).
- In Section 3.1, you state that the time-frequency dilation and distortion makes parameter estimation difficult and you therefor propose FRFT. Before immediately jumping into the math, can you offer some insight as to why FRFT might solve this issue and yield a better result?
- As you are introducing various initialisms in Section 3, can you please redefine the full phrases at least once for clarity?
- Section 3.2 clearly outlines that window length selection is critical for STFT and you propose SST to improve this limitation. Can you provide additional context for what expected performance enhancements you expect this to yield and why you expect it to perform better? The section abruptly ends without any real conclusion about the expected / actual impact of the proposed change.
- In sections 4.1, 4.2, and 4.3, you very quickly introduce several estimators without very much discussion about how they are formulated. It is possible that it is just obvious and I don't have the insight to see it, in which case ignore this comment. If that is not the case, however, I would like to see significantly more discussion about how these estimators were derived from the expressions you cite throughout the section, or a citation to another body of work that describes them in detail, if appropriate.
- Figure 3 seems out of place because it is not supported by the text at all. Please consider either adding a paragraph discussing the processing flow or at least a detailed caption.
- The section 5 title and subsections 5.1 and 5.2 titles are too long and also unclear, please consider more concise and descriptive titles.
- Figure 4 is not well supported by the text, before immediately jumping into an analysis of the results please consider describing the configuration depicted in each plot. I recognize that the information is technically available in the caption but it is difficult to parse and can easily be solved with a couple of sentences.
- Figures 5, 6, and 7 are likewise poorly supported by text. The explanation provided in section 5.2 is disorganized and poorly written, so it is difficult to understand what is happening without multiple re-reads. This whole section needs to be carefully re-written.
- Throughout the manuscript, you provide many of the setup parameters in the text. You may consider instead placing all of these values in tables for clarity.
- Section 6.1 is likewise disorganized, quickly introducing random information without clearly explaining that content and importance of each figure. This section also needs a re-write.
- Section 6 needs an introduction that clearly states what kind of simulation environment you are building, why you are building it, and what insight you expect to gain (or did gain) by doing so. Similarly, this introduction also needs to introduce the fact that you are collecting experimental data and briefly describe introduce any relevant details. As it stands, Section 6.2 is a total surprise and immediately starts delivering experimental results without any significant context.
- In section 6.2, you state that you removed the "body" in Figure 10 (b); can you provide some context on exactly how you did that?
- Section 6 ends without any real discussion about the results. What is the quantitative difference in the proposed method vs the standard STFT approach (10 (c) vs 10 (d))? Why did you simulate a 95 GHz radar when your experimental configuration uses a 674 MHz radar? What are the quantitative differences in simulation? Why is the simulation configuration different than the experimental differences? What implications does this have on your results? If you intend to present these results as qualitative, then they need to supported by sufficient discussion. If they are supposed to be quantitative, then there needs to be some sort of representative numerical comparison. In the current state, the manuscript doesn't clearly articulate what the results are or why they are compelling (if at all). In the introduction, you state that you are applying FRFT-FSST to address the limitations of existing approaches, but I do not see a compelling discussion of what these limitations are, how they are mitigated by the proposed approach, or the quantitative / qualitative improvements. Sections 5 and 6 are disorganized and lacking in context and clarity. The initial simulation and experimental results look promising but these two sections require significant revisions to improve clarity.
Author Response
We are very grateful to the Reviewers, the Associate Editor, and the Editor in Chief for their great efforts involved in handling and reviewing our paper. The valuable Reviewers’ Comments are truly helpful for us to improve the quality of the submitted manuscript. We have carefully revised the manuscript according to these suggestions. This report summarizes our point-to-point response to the Reviewers’ Comments. Please kindly note that our replies begin with a bold “Reply” and end with a black square.

Reviewer 2 Report
The topic of this paper is interesting, and the workload is sufficient to present the signal detection and parameter estimation using the time-frequency analysis. Numerical results obtained from simulation and measurement data are provided to verify the effectiveness. I have the following suggestions:
1) Classical signal detection method, e.g., GLRT, Rao, Wald tests should be added into the introduction part to enrich the content. Moreover, lots of new detectors [1-2] can also be added into the introduction part, as they are proven to be effective for signal detection in low SNR.
[1] "Target Detection Within Nonhomogeneous Clutter Via Total Bregman Divergence-Based Matrix Information Geometry Detectors," in IEEE Transactions on Signal Processing, vol. 69, pp. 4326-4340, 2021, doi: 10.1109/TSP.2021.3095725.
[2] "Persymmetric Subspace Detectors With Multiple Observations in Homogeneous Environments," in IEEE Transactions on Aerospace and Electronic Systems, vol. 56, no. 4, pp. 3276-3284, 2020.
2) It will be better to consider other time-frequency features for micro-Doppler signal detection, such as WVD, CWD .
Author Response

(The authors gave the same response as above.)

Reviewer 3 Report
This paper presents Rotor UAV's Micro-Doppler Signal Detection and Parameter Estimation Based on FRFT-FSST. The work looks interesting and would be interesting to the experts in signal processing research fields. The manuscript has to be revised before publish as a Journal paper. Some comments to improve the manuscript are as below; 1. In title authors mention about "parameter estimation". Authors clearly mention parameters of what. 2. This work is related to drone related signal processing. Some works in the literature have done drone sound signal processing. For example; Audio-processing-based Human Detection at Disaster Sites with Unmanned Aerial Vehicle (same authors may have published related papers), Development of microphone-array-embedded UAV for search and rescue task. I recommend authors discuss these kind of works in the literature to broaden the literature. 3. Theory part of the paper is written well. But, the experimental part is weak. I recommend authors to further add results of experiments with real machines. As you may know that as motors changes results would be different. 4. As a overall comment, the paper is well written. But some minor issue in the sentences. Specially, please recheck the sentences in the conclusion.Author Response
We are very grateful to the Reviewers, the Associate Editor, and the Editor in Chief for their great efforts involved in handling and reviewing our paper. The valuable Reviewers’ Comments are truly helpful for us to improve the quality of the submitted manuscript. We have carefully revised the manuscript according to these suggestions. This report summarizes our point-to-point response to the Reviewers’ Comments. Please kindly note that our replies begin with a bold “Reply” and end with a black square.

Reviewer 4 Report
1) I am not satisfied with the current review of the literature provided in the Introduction. On the one hand, there are other important and quite recent references about UAV's micro-Doppler signal detection that are missing and deserve to be taken into account:
- Coluccia, A et al: Detection and Classification of Multirotor Drones in Radar Sensor Networks: A Review. Sensors, 2020;
- A. Huizing et al "Deep Learning for Classification of Mini-UAVs Using Micro-Doppler Spectrograms in Cognitive Radar," IEEE Aerospace and Electronic Systems Magazine, 2019;
- Y. Sun et al: "Micro-Doppler Signature-Based Detection, Classification, and Localization of Small UAV With Long Short-Term Memory Neural Network," in IEEE Transactions on Geoscience and Remote Sensing, Aug. 2021.
On the other hand, the authors do not properly discuss the main difficulties related to the adoption of micro-Doppler based detection techniques when applied to identify the rotors of UAVs. In the literature, there is a vast amount of work discussing the main challenges (es., distinguishing drone from birds, accurately considering the different types of rotors, ...). In this respect, authors should provide an (even concise) discussion to put the focus on such important aspects, justified by pointers to the existing literature:
- Y. D. Zhang et al: "Enhanced Micro-Doppler Feature Analysis for Drone Detection," 2021 IEEE Radar Conference (RadarConf21), 2021;
- A. Schumann et al, Drone vs. Bird Detection: Deep Learning Algorithms and Results from a Grand Challenge. Sensors, 2021;
- J. Wang et al, "Counter-Unmanned Aircraft System(s) (C-UAS): State of the Art, Challenges, and Future Trends," in IEEE Aerospace and Electronic Systems Magazine, March 2021.
2) The specific choice of the Kernel function in eq. (13) needs to be explained and justified. Are there alternative though still valid Kernel functions that can be potentially used? If so, what are they?
3) Please provide an explicit definition for the Gaussian function g(t) after eq. (18). All the symbols need to be properly defined.
4) How are the threshold coefficients after eq. (25) chosen in practice?
5) A table summarizing the main simulation parameters adopted in Sec. 6 would be helpful.
6) The size of the labels as well as of the axes need to be improved in Fig. 10(b).
7) A complexity analysis of the proposed approach in Sec. 6 is missing. It is important to quantify the computational burden in order to highlight potential trade-offs between the accuracy achieved and the cost involved in the processing.
Author Response

(The authors gave the same response as above.)

Round 2
Reviewer 1 Report
- I gave minor feedback on the abstract. This issue has been mostly resolved in the revised version. In the first sentence, I would replace the comma before "however" with a semicolon, and end the sentence after "rotor parameter estimation". No further revisions are necessary.
- I gave minor feedback on the sentence structure and content of the introduction, specifically asking for context of the proposed method with respect to existing literature. This issue has been mostly addressed in the revised version. There are some minor typographical mistakes in the revised version that can be trivially fixed.
- I gave minor feedback on the formatting of equations, which appear to have been fixed in the revised version.
- I requested some insight on why the FRFT might be expected to yield better results. The authors added a brief explanation and some corresponding references to answer my question.
- I made minor comments on initialisms later in the paper which have been addressed in the revised version.
- I requested additional clarity on the window length selection, which the authors addressed in the first and last paragraphs of section 3.2 in the revised version.
- I requested additional discussion on the equations in sections 4.1 - 4.3. The authors have added sufficient discussion and clarification in the revised manuscript.
- I requested additional discussion of Figure 3, which the authors added in the revised version.
- I suggested that the titles and subtitles in Section 5 be reduced, which the authors did.
- I requested additional clarity on Figure 4, which the authors now provide in Table 1.
- I gave similar feedback on Figures 5, 6, and 7, which the authors addressed similarly.
- I requested a re-write of Sections 5.2 and 6.1, which the authors provide in the revised manuscript.
- I requested clarification on the removal of the aircraft body, which the authors clarified with a short paragraph and some citations.
- I requested additional discussion on the quantitative results at the end of Section 6, which the others provide in the revised manuscript.
- I requested clarification on the significant difference between the simulated and experimental configurations. The authors clarified the goals of each test and why the configurations were different.
- I requested additional concluding discussion about the impact of the proposed approach. The authors have added a nice discussion of this in the conclusion.
All of the suggestions and requests that I made in my previous review appear to have been sufficiently addressed by the authors in the revised manuscript. There are still some minor formatting and typographical problems, but I trust that these can be resolved by the authors without rigorous review. After a final formatting revision, I believe this manuscript will be suitable for publication. Thank you for your contributions and professional response to my feedback, as well as your patience while I prepared this review.
Reviewer 2 Report
It has been fully revised.
Reviewer 3 Report
I appreciate the revision by the authors.
Reviewer 4 Report
The authors correctly addressed all my comments.